# BOtied: Multi-objective Bayesian optimization with tied multivariate ranks

## Abstract

Many scientific and industrial applications require the joint optimization of multiple, potentially competing objectives. Multi-objective Bayesian optimization (MOBO) is a sample-efficient framework for identifying Pareto-optimal solutions. At the heart of MOBO is the acquisition function, which determines the next candidate to evaluate by navigating the best compromises among the objectives. Multi-objective acquisition functions that rely on box decomposition of the objective space, such as the expected hypervolume improvement (EHVI) and entropy search, scale poorly to a large number of objectives. We begin by showing a natural connection between non-dominated solutions and the highest multivariate rank, which coincides with the outermost level line of the joint cumulative distribution function (CDF). Motivated by this link, we propose the CDF indicator, a Pareto-compliant metric for evaluating the quality of approximate Pareto sets that complements the popular hypervolume indicator. We then propose an acquisition function based on the CDF indicator, called BOtied. BOtied can be implemented efficiently with copulas, a statistical tool for modeling complex, high-dimensional distributions. We benchmark BOtied against common acquisition functions, including EHVI, entropy search, and random scalarization, in a series of synthetic and real-data experiments. BOtied performs on par with the baselines across datasets and metrics while being computationally efficient.

## 1 Introduction

Bayesian optimization (BO) has demonstrated promise in a variety of scientific and industrial domains where the goal is to optimize an expensive black-box function using a limited number of potentially noisy function evaluations (Romero et al., 2013; Calandra et al., 2016; Kusne et al., 2020; Shields et al., 2021; Zuo et al., 2021; Bellamy et al., 2022; Khan et al., 2023; Park et al., 2022). In BO, we fit a probabilistic surrogate model on the available observations so far. Based on the model, the acquisition function determines the next candidate to evaluate by balancing exploration (evaluating highly uncertain candidates) with exploitation (evaluating designs believed to maximize the objective). Often, applications call for the joint optimization of multiple, potentially competing objectives (Marler & Arora, 2004; Jain et al., 2017; Tagasovska et al., 2022). Unlike in single-objective settings, a single optimal solution may not exist and we must identify a set of solutions that represents the best compromises among the multiple objectives. The acquisition function in multi-objective Bayesian optimization (MOBO) navigates these trade-offs as it guides the optimization toward regions of interest.

A computationally attractive approach to MOBO scalarizes the objectives with random preference weights (Knowles, 2006; Paria et al., 2020) and applies a single-objective acquisition function. The distribution of the weights, however, may be insufficient to encourage exploration when there are many objectives with unknown scales. Alternatively, we may address the multiple objectives directly by seeking improvement on a set-based performance metric, such as the hypervolume (HV) indicator (Emmerich, 2005; Emmerich et al., 2011; Daulton et al., 2020; 2021) or the R2 indicator (Deutz et al., 2019a;b). Improvement-based acquisition functions are sensitive to the rescaling of the objectives, which may carry drastically different natural units. In particular, computing the HV has time complexity that is super-polynomial in the number of objectives, because it entails computing the volume of an irregular polytope (Yang et al., 2019). Despite the efficiency improvement achieved by

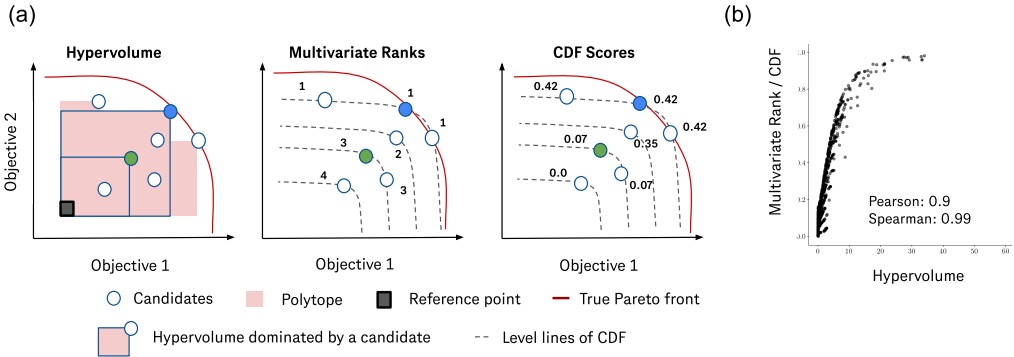

Figure 1: (a) Conceptual summary of BOTIED: Here, the blue candidate is predicted to dominate green with respect to both objectives. The HV indicator is consistent with this ordering; the area of the box bounded by the blue candidate is bigger than that bounded by the green. Multivariate ranks and CDF scores, used in BOTIED, also favor the blue. (b) The CDF scores closely trace HV.

box decomposition algorithms (Dächert et al., 2017; Yang et al., 2019), HV computation remains slow when the number of objectives exceeds 4. Another class of acquisition strategies is entropy search, which focuses on maximizing the information gain from the next observation (Hernández-Lobato et al., 2014; 2016b; Shah & Ghahramani, 2015; Belakaria et al., 2019; Hoffman & Ghahramani, 2015; Tu et al., 2022). Computing entropy-based acquisition functions also involves computing high-dimensional definite integrals, this time of an $M$-dimensional multivariate Gaussian, where $M$ is the number of objectives. They are commonly implemented in box decompositions as well, but are even more costly to evaluate than HV.

Many bona fide MO acquisition functions without scalarization, such as EHVI or entropy searches, involve high-dimensional integrals and scale poorly with increasing numbers of objectives. EHVI and random scalarization are sensitive to non-informative transformations of the objectives, such as rescaling of one objective relative to another or monotonic transformations of individual objectives. To address these challenges, we propose BOTIED[1], a novel acquisition function based on multivariate ranks. We show that BOTIED has the desirable property of being invariant to relative rescaling or monotonic transformations of the objectives. While it maintains the multivariate structure of the objective space, its implementation has highly favorable time complexity and we report wall-clock time competitive with random scalarization.

In Fig. 1(a), we present the intuition behind multivariate ranks. Consider a maximization setup over two objectives where we seek to identify solutions on the true Pareto frontier (red curves), hypothetical and inaccessible to us. Suppose we have many candidates, represented as circular posterior blobs in the objective space, where the posteriors have been inferred from our probabilistic surrogate model. For simplicity, assume the posterior widths (uncertainties) are comparable among the candidates. Let us consider each candidate individually. How do we estimate each candidate's proximity to the true Pareto frontier? Our surrogate model predicts the candidate shaded in blue to have high values in both objectives and, unbeknownst to us, it happens to lie on the true Pareto front. On the other hand, the candidate shaded in green is predicted to be strictly dominated by the blue counterpart. The areas of regions bounded from above by the candidates corroborate this ordering, as shown in the leftmost panel; the HV dominated by the blue candidate is bigger than that of the green. Alternatively, we can compute multivariate ranks of the candidates (middle panel). Consistent with the HV ordering, the blue candidate is ranked higher, at 1, than the green candidate, at 3. Note that, due to orthogonality, there may be a *tie* among the candidates.

Ranking in high dimensions is not a trivial task, as there is no natural ordering in Euclidean spaces when $M \geq 2$. To compute multivariate ranks, we propose to use the (joint) cumulative distribution function (CDF) defined as the probability of a sample having greater function value than other candidates, $F_Y(y) = P(f(X) \leq y)$, where $y = f(x)$. The gray dashed lines indicate the level lines of the CDF. The level line at $F(\cdot) = 1$ is the Pareto frontier estimated by our CDF. As Fig. 1(b) shows, the CDF scores themselves closely trace HV as well. Leveraging ranks has been explored in the many-objective literature such as Kukkonen & Lampinen (2007), but limited to computing

---

[1]The name choice stems from non-dominated candidates considered as "tied".

the ranks of individual objectives and combining them post-hoc by a simple aggregation function (min, max, average) in order to obtain the overall fitness values. Similarly, Picheny et al. (2019) propose Ordinal BO which accounts for the rankings of both solutions and objective values, but their method is only suitable for discrete spaces. Differently, here, we propose a joint, multivariate ranking approach for continuous data.

Motivated by the natural interpretation of multivariate ranks as a multi-objective indicator, we make the following contributions: (i) We propose a new Pareto-compliant performance criterion, the CDF indicator (Sec. 2); (ii) We propose a scalable and robust acquisition function based on the multirank, BOTIED (Sec. 3); (iii) We release a benchmark dataset that explores an ideal case for BOTIED ranking, in which we can specify the correct data-generating model when fitting the CDF of the objectives (Sec. 4). The dataset probes particular dependency structures in the objectives and opens the door to incorporating domain knowledge.

## 2 BACKGROUND

### 2.1 BAYESIAN OPTIMIZATION

Bayesian optimization (BO) is a popular technique for sample-efficient black-box optimization (see Shahriari et al., 2015; Frazier, 2018, for a review). In a single-objective setting, suppose our objective $f : \mathfrak{X} \to \mathbb{R}$ is a black-box function of the design space $\mathfrak{X}$ that is expensive to evaluate. Our goal is to efficiently identify a design $\boldsymbol{x}^\star \in \mathfrak{X}$ maximizing[2] $f$. BO leverages two tools, a probabilistic surrogate model and a utility function, to trade off exploration (evaluating highly uncertain designs) and exploitation (evaluating designs believed to maximize $f$) in a principled manner.

For each iteration $t \in \mathbb{N}$, we have a dataset $\mathcal{D}_t = \{(\boldsymbol{x}^{(1)}, y^{(1)}), \cdots, (\boldsymbol{x}^{(N_t)}, y^{(N_t)})\} \in \mathscr{D}_t$, where each $y^{(n)}$ is a noisy observation of $f(\boldsymbol{x}^{(n)})$. First, the probabilistic model $\hat{f} : \mathfrak{X} \to \mathbb{R}$ infers the posterior distribution $p(f|\mathcal{D}_t)$, quantifying the plausibility of surrogate objectives $\hat{f} \in \mathcal{F}$. Next, we introduce a utility function $u : \mathfrak{X} \times \mathcal{F} \times \mathscr{D}_t :\to \mathbb{R}$. The acquisition function $a(\boldsymbol{x})$ is simply the expected utility of $\boldsymbol{x}$ w.r.t. our current belief about $f$,

$$a(\boldsymbol{x}) = \int u(\boldsymbol{x}, \hat{f}, \mathcal{D}_t) p(\hat{f}|\mathcal{D}_t) d\hat{f}. \tag{2.1}$$

For example, we obtain the expected improvement (EI) acquisition function if we take $u_{\mathrm{EI}}(\boldsymbol{x}, \hat{f}, \mathcal{D}) = [\hat{f}(\boldsymbol{x}) - \max_{(\boldsymbol{x}', y') \in \mathcal{D}} y']_+$, where $[\cdot]_+ = \max(\cdot, 0)$ (Močkus, 1975; Jones et al., 1998). Often the integral is approximated by Monte Carlo (MC) with posterior samples $\hat{f}^{(j)} \sim p(f|\mathcal{D}_t)$. We select a maximizer of $a$ as the new design, evaluate $f(a)$, and append the observation to the dataset. The surrogate is then refit on the augmented dataset and the procedure repeats.

### 2.2 MULTI-OBJECTIVE OPTIMIZATION

When there are multiple objectives of interest, a single best design may not exist. Suppose there are $M$ objectives, $f : \mathfrak{X} \to \mathbb{R}^M$. The goal of multi-objective BO is to identify the set of *Pareto-optimal* solutions such that improving one objective within the set leads to worsening another. We say that $\boldsymbol{x}$ dominates $\boldsymbol{x}'$, or $f(\boldsymbol{x}) \succ f(\boldsymbol{x}')$, if $f_m(\boldsymbol{x}) \geq f_m(\boldsymbol{x}')$ for all $m \in \{1, \ldots, M\}$ and $f_m(\boldsymbol{x}) > f_m(\boldsymbol{x}')$ for some $m$. The set of *non-dominated* solutions $\mathscr{X}^*$ is defined in terms of the Pareto frontier (PF) $\mathcal{P}^*$,

$$\mathscr{X}^\star = \{\boldsymbol{x} : f(\boldsymbol{x}) \in \mathcal{P}^\star\}, \quad \text{where } \mathcal{P}^\star = \{f(\boldsymbol{x}) : \boldsymbol{x} \in \mathfrak{X}, \nexists \boldsymbol{x}' \in \mathfrak{X} \text{ s.t. } f(\boldsymbol{x}') \succ f(\boldsymbol{x})\}. \tag{2.2}$$

Multi-objective BO algorithms typically aim to identify a finite subset of $\mathscr{X}^\star$, which may be infinite, within a reasonable number of iterations.

**Hypervolume** One way to measure the quality of an approximate PF $\mathcal{P}$ is to compute the hypervolume (HV) $\mathrm{HV}(\mathcal{P}|\boldsymbol{r}_{\mathrm{ref}})$ of the polytope bounded from above by $\mathcal{P}$ and from below by $\boldsymbol{r}_{\mathrm{ref}}$, where

---

[2]For simplicity, we define the task as maximization in this paper without loss of generality. For minimizing $f$, we can negate $f$, for instance.

$\boldsymbol{r}_{\mathrm{ref}} \in \mathbb{R}^M$ is a user-specified *reference point*. More specifically, the HV indicator for a set $A$ is

$$I_{\mathrm{HV}}(A) = \int_{\mathbb{R}^M} \mathbb{I}[\boldsymbol{r}_{\mathrm{ref}} \preceq \boldsymbol{y} \preceq A] d\boldsymbol{y}. \tag{2.3}$$

We obtain the expected hypervolume improvement (EHVI) acquisition function if we take

$$u_{\mathrm{EHVI}}(\boldsymbol{x}, \hat{f}, \mathcal{D}) = \mathrm{HVI}(\mathcal{P}', \mathcal{P}|\boldsymbol{r}_{\mathrm{ref}}) = [I_{\mathrm{HV}}(\mathcal{P}'|\boldsymbol{r}_{\mathrm{ref}}) - I_{\mathrm{HV}}(\mathcal{P}|\boldsymbol{r}_{\mathrm{ref}})]_+, \tag{2.4}$$

where $\mathcal{P}' = \mathcal{P} \cup \{\hat{f}(\boldsymbol{x})\}$ (Emmerich, 2005; Emmerich et al., 2011).

**Noisy observations** In the noiseless setting, the observed baseline PF is the true baseline PF, i.e. $\mathcal{P}_t = \{\boldsymbol{y} : \boldsymbol{y} \in \mathcal{Y}_t, \not\exists \boldsymbol{y}' \in \mathcal{Y}_t \ s.t. \ \boldsymbol{y}' \succ \boldsymbol{y}\}$, where $\mathcal{Y}_t := \{\boldsymbol{y}^{(n)}\}_{n=1}^{N_t}$. This does not, however, hold in many practical applications, where measurements carry noise. For instance, given a zero-mean Gaussian measurement process with noise covariance $\Sigma$, the feedback for a candidate $\boldsymbol{x}$ is $\boldsymbol{y} \sim \mathcal{N}(f(\boldsymbol{x}), \Sigma)$, not $f(\boldsymbol{x})$ itself. The *noisy* expected hypervolume improvement (NEHVI) acquisition function marginalizes over the surrogate posterior at the previously observed points $\mathcal{X}_t = \{\boldsymbol{x}^{(n)}\}_{n=1}^{N_t}$,

$$u_{\mathrm{NEHVI}}(\boldsymbol{x}, \hat{f}, \mathcal{D}) = \mathrm{HVI}(\hat{\mathcal{P}}_t', \hat{\mathcal{P}}_t|\boldsymbol{r}_{\mathrm{ref}}), \tag{2.5}$$

where $\hat{\mathcal{P}}_t = \{\hat{f}(\boldsymbol{x}) \ : \ \boldsymbol{x} \in \mathcal{X}_t, \ \not\exists \boldsymbol{x}' \in \mathcal{X}_t \ s.t. \ \hat{f}(\boldsymbol{x}') \succ \hat{f}(\boldsymbol{x})\}$ and $\hat{\mathcal{P}}' = \hat{\mathcal{P}} \cup \{\hat{f}(\boldsymbol{x})\}$ (Daulton et al., 2021).

## 3 MULTI-OBJECTIVE BO WITH TIED MULTIVARIATE RANKS

In MOBO, it is common to evaluate the quality of an approximate Pareto set $\mathcal{X}$ by computing its distance from the optimal Pareto set $\mathcal{X}^*$ in the objective space, or $d(f(\mathcal{X}), f(\mathcal{X}^*))$. The distance metric $d : 2^{\mathcal{Y}} \times 2^{\mathcal{Y}} \to \mathbb{R}$ quantifies the difference between the sets of objectives, where $2^{\mathcal{Y}}$ is the power set of the objective space $\mathcal{Y}$. Existing work in MOBO mainly focuses on the difference in HV, or HVI. One advantage of HV is its sensitivity to any type of improvement, i.e., whenever an approximation set $A$ dominates another approximation set $B$, then the measure yields a strictly better quality value for the former than for the latter set (Zitzler et al., 2003). Although HV is the most common metric of choice in MOBO, it suffers from sensitivity to transformations of the objectives and scales super-polynomially with the number of objectives, which hinders its practical value. An alternative approach is to use distance-based indicators (Miranda & Von Zuben, 2016; Wang et al., 2016b) that assign scores for the solutions based on a signed distance from each point to an estimated Pareto front, which is again computationally expensive. On the other hand, multivariate ranks inherently approximate the distance to the Pareto front as explained in Sec. 1, Fig. 1. Methods such as (Miranda & Von Zuben, 2016; Wang et al., 2016b) can be considered as precursors to our multivariate indicator.

In the following, the *(weak) Pareto-dominance* relation is used as a preference relation $\succcurlyeq$ on the search space $X$ indicating that a solution $x$ is at least as good as a solution $y$ $(x \succcurlyeq y)$ if and only if $\forall 1 \le i \le M : f_i(x) \ge f_i(y)$. This relation can be canonically extended to sets of solutions where a set $A \subseteq X$ weakly dominates a set $B \subseteq X (A \succcurlyeq B)$ iff $\forall y \in B \ \exists x \in A : x \succcurlyeq y$ (Zitzler et al., 2003). Given the preference relation, we consider the optimization goal of identifying a set of solutions that approximates the set of Pareto-optimal solutions and ideally this set is not strictly dominated by any other approximation set.

Since the generalized weak Pareto dominance relation defines only a partial order on $\mathcal{Y}$, there may be incomparable sets in $\mathcal{Y}$ which may cause difficulties with respect to search and performance assessment. These difficulties become more serious as $M$ increases (see Fonseca et al. (2005) for details). One way to circumvent this problem is to define a total order on $\mathcal{Y}$ which guarantees that any two objective vector sets are mutually comparable. To this end, quality indicators have been introduced that assign, in the simplest case, each approximation set a real number, i.e., a (unary) indicator $I$ is a function $I : \mathcal{Y} \to \mathbb{R}$ (Zitzler et al., 2003). One important feature an indicator should have is *Pareto compliance* Fonseca et al. (2005), i.e., it must not contradict the order induced by the Pareto dominance relation.

In particular, this means that whenever $A \succcurlyeq B \land B \not\succcurlyeq A$, then the indicator value of A must not be worse than the indicator value of B. A stricter version of compliance would be to require that the indicator value of A is strictly better than the indicator value of B (if better means a higher indicator value):

$$A \succcurlyeq B \land B \not\succcurlyeq A \Rightarrow I(A) > I(B). \tag{3.1}$$

## 3.1 CDF INDICATOR

Here we suggest using the CDF as an indicator for measuring the quality of Pareto approximations.

**Definition 1** (Cumulative distribution function). *The CDF of a real-valued random variable $Y$ is the function given by:*

$$F_Y(y) = P(Y \leq y) = \int_{-\infty}^{y} p_Y(t)dt. \tag{3.2}$$

*i.e. it represents the probability that the r.v. $Y$ takes on a value less than or equal to $y$.*

For more then two variables, the joint CDF is given by:

$$F_{Y_1,\ldots,Y_M} = P(Y_1 \leq y_1,\ldots,Y_M \leq y_m) = \int_{(-\infty,\ldots,-\infty)}^{(y_1,\ldots,y_M)} p_Y(\mathbf{s})ds. \tag{3.3}$$

**Properties of the CDF** Every multivariate CDF is (i) monotonically non-decreasing for each of its variables, (ii) right-continuous in each of its variables and (iii) $0 \leq F_{Y_1,\ldots,Y_M}(y_1,\ldots,y_m) \leq 1$. The monotonically non-decreasing property means that $F_{\mathbf{Y}}(a_1,\ldots,a_M) \geq F_{\mathbf{Y}}(b_1,\ldots,b_M)$ whenever $a_1 \geq b_1,\ldots,a_K \geq b_M$. We leverage these properties to define our CDF indicator.

**Definition 2** (CDF Indicator). *The CDF indicator $I_F$ is defined as the maximum multivariate rank*

$$I_{F_{\mathbf{Y}}}(A) := \max_{y \in A} F_{\mathbf{Y}}(y), \tag{3.4}$$

*where $A$ is an approximation set in $\Omega$.*

Next we show that this indicator is compliant with the concept of Pareto dominance.

**Theorem 1** (Pareto compliance). *For any arbitrary approximation sets $A \in \Omega$ and $B \in \Omega$, it holds*

$$A \succcurlyeq B \wedge B \not\succcurlyeq A \Rightarrow I_F(A) \geq I_F(B). \tag{3.5}$$

The proof can be found in Appendix A.

**Remark 1.** *Note that $I_{F_{\mathbf{Y}}}$ in Eq. 3.4 only depends on the best element in the $F_{\mathbf{Y}}$ rank ordering. One consequence of this is that $I_{F_{\mathbf{Y}}}$ does not discriminate sets with the same best element.*

## 3.2 ESTIMATION OF THE CDF INDICATOR WITH COPULAS

Computing a multivariate joint distribution $F_{\mathbf{Y}}$ is a challenging task. A naive approach involves estimating the multivariate density function and then computing the integral, which is computationally intensive. We turn to *copulas* Nelsen (2007); Bedford & Cooke (2002a), statistical tool for flexible density estimation in higher dimensions.

**Theorem 2** (Sklar's theorem Sklar (1959)). *The continuous random vector $Y = (Y_1,\ldots,Y_M)$ has a joint distribution $F$ and marginal distributions $Y_1,\ldots,F_M$ if and only if there exist a unique copula $C$, which is the joint distribution of $U = (U_1,\ldots,U_M) = F_1(Y_1),\ldots,F_d(Y_M)$.*

From Sklar's theorem, we note that a copula is a multivariate distribution function $C : [0,1]^M \to [0,1]$ that joins (couples) uniform marginal distributions:

$$F(y_1,\ldots,y_M) = C\left(F_1(y_1),\ldots,F_d(y_M)\right). \tag{3.6}$$

By computing the copula function, we also obtain access to the multivariate CDF and, by construction, to the multivariate ranking.

It is important to note that, to be able to estimate a copula, we need to transform the variables of interest to uniform marginals. We do so by the so-called *probability integral transform (PIT)* of the marginals.

**Definition 3** (Probability integral transform). *PIT of a random variable $Y$ with distribution $F_Y$ is the random variable $U = F_Y(y)$, which is uniformly distributed: $U \sim \text{Unif}([0,1])$.*

The benefits of using copulas as estimators for the CDF indicator are threefold: (i) Scalability and flexible estimation in higher dimensional objective spaces, (ii) Scale invariance wrt different objectives, (iii) Invariance under monotonic transformations of the objectives. These three properties suggest that our indicator is more robust than the widely used HV indicator, as we will empirically show in the following section. Sklar's theorem, namely the requirement of uniform marginals, immediately implies the following corollary which characterizes the invariance of the CDF indicator to different scales.

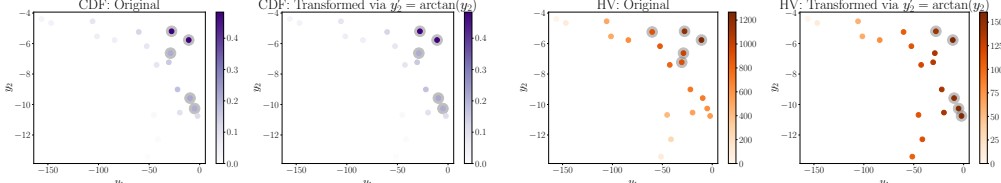

Figure 2: The CDF indicator is invariant to arbitrary monotonic transformations of the objectives (here transforming property $y_2$ via arctan), whereas the HV indicator is highly sensitive to them. The color gradient corresponds to the value of the indicator at each data point. Gray circles are overlaid on the five solutions with the top indicator scores. CDF chooses the same five solutions, but HV prefers solutions with high $y_1$ after $y_2$ becomes squashed via arctan.

**Corollary 1** (Scale invariance). *A copula based estimator for the CDF indicator is scale-invariant.*

**Corollary 2** (Invariance under monotonic transformations). *Let $Y_1, Y_2$ be continuous random variables with copula $C_{Y_1,Y_2}$. If $\alpha, \beta : \mathbb{R} \to \mathbb{R}$ are strictly increasing functions, then:*

$$C_{\alpha(Y_1),\beta(Y_2)} = C_{Y_1,Y_2} \tag{3.7}$$

*where $C_{\alpha(Y_1),\beta(Y_2)}$ is the copula function corresponding to variables $\alpha(Y_1)$ and $\beta(Y_2)$.*

Corollary 1 follows from the PIT required for copula estimation. The proof for invariance under monotonic transformations based on Haugh (2016) can be found in Sec. A.2 and, without loss of generality, can be extended to more than two dimensions. We empirically validate the robustness properties of the copula-based estimator in Fig. 2.

The scale-invariance benefit of the copula transformation has been exploited in other works in the context of optimization. Namely, Binois et al. (2020) leverage the copula space when finding a solution to their game-theoretic MO approach, while Eriksson & Poloczek (2021) use Gaussian copula transformations to magnify values at the end of the observed ranges (suitable for their constraint-oriented objective function). A detailed overview and positioning of our multivariate approach with regards to related work can be found in Appendix D, Table 2.

### 3.3 CDF-BASED ACQUISITION FUNCTION: BOTIED

Suppose we fit a CDF on $\boldsymbol{y}^{(1)}, \boldsymbol{y}^{(2)}, \ldots, \boldsymbol{y}^{(N_t)}$, the $N_t$ measurements acquired so far. Denote the resulting CDF as $\hat{F}(\cdot; \mathcal{D}_t)$, where we have made explicit the dependence on the dataset up to time $t$. The utility function of our BOTIED acquisition function is as follows:

$$u(\mathbf{x}, \hat{f}, \mathcal{D}_t) = \hat{F}(\hat{f}(\mathbf{x}); \mathcal{D}_t). \tag{3.8}$$

### 3.4 ESTIMATING HIGH-DIMENSIONAL CDFS WITH VINE COPULAS

As with the CDF indicator, our CDF-based acquisition function has an efficient implementation that makes use of copulas. For a more complete, self-contained description of how a copula acquisition function fits within a single round of MOBO, we include Algorithm 1 in Appendix E. The copula score can be found between lines 10 and 13.

A copula can be modeled following a parametric family depending on the shape of the dependence structure (e.g., Clayton copula with lower tail dependence, Gumbel copula with upper tail, Gaussian, no tail dependence but full covariance matrix). For additional flexibility and scalability, Bedford & Cooke (2002b) has proposed *vine copulas*, a pair-copula construction that allows the factorization of any joint distribution into bivariate copulas. In this construction, the copula estimation problem decomposes into two steps. First, we specify a graphical model, structure called *vine* consisting of $M(M-1)/2$ number of trees. Second, we choose a parametric or nonparametric estimator for each edge in the tree representing a bivariate copula. Aas (2016) propose efficient algorithms to organize the trees. For completeness, we add an example decomposition in Appendix C as well as the steps involved in fitting a copula. For more details on these algorithms and convergence guarantees, please see Bedford & Cooke (2002b) and references therein.

## 4 EMPIRICAL RESULTS

**Experimental Setup** To empirically evaluate the sample efficiency of BOTIED, we execute simulated BO rounds on a variety of problems. See Appendix F for more details about our experiments. Our codebase is available in the supplementary material and at `anonymous.link`.

**Metrics** We use the HV indicator presented in Sec. 3, the standard evaluation metric for MOBO, as well as our CDF indicator. We rely on efficient algorithms for HV computation based on hyper-cell decomposition as described in (Fonseca et al., 2006; Ishibuchi et al., 2011) and implemented in `BoTorch` (Balandat et al., 2020).

**Baselines** We compare BOTIED with popular acquisition functions. We assume noisy function evaluations, so implement noisy versions of all the acquisition functions. The baseline acquisition strategies include *NEHVI* (noisy EHVI) Daulton et al. (2020) described in Eq. 2.5; *NParEGO* (noisy ParEGO) Knowles (2006) which uses random augmented Chebyshev scalarization and noisy expected improvement; multiple entropy search baselines, Joint entropy search (JES) (Hvarfner et al., 2022) , Maximum Entropy Search (MES) (Belakaria et al., 2019) and Predictive Entropy Search (PES) (Hernández-Lobato et al., 2016a), the difference between the three being the whether the entropy of inputs or objectives or both is being considered. Finally, we add the *random* selection baseline. For BOTIED we have two implementations, v1 and v2, both based on the joint CDF estimation with the only difference being the method of incorporating the variance from the Monte Carlo (MC) predictive posterior samples, either fitting the copula on all of them (v1) or on the means (v2). The algorithms for both versions can be found in Appendix E, Algorithm 1.

**Datasets** As a numerical testbed, we begin with toy test functions commonly used as BO benchmarks: Branin-Currin (Daulton et al., 2022a) ($d = 2$, $M = 2$) and DTLZ (Deb & Gupta, 2005) ($d = 9$, $M \in \{4, 6, 8\}$). The Penicillin test function (Liang & Lai, 2021) ($d = 7$, $M = 3$) simulates the penicillin yield, time to production, and undesired byproduct for various parameters of the production process. All of these tasks allow for a direct evaluation of $f$.

To emulate a real-world drug design setup, we modify the permeability dataset Caco-2 Wang et al. (2016a) from the Therapeutics Data Commons database (Huang et al., 2021; 2022). Permeability is a key property in the absorption, distribution, metabolism, and excretion (ADME) profile of drugs. The Caco-2 dataset consists of 906 drug molecules annotated with experimentally measured rates of passing through a human colon epithelial cancer cell line. We represent each molecule as a concatenation of fingerprint and fragment feature vectors, known as fragprints (Thawani et al., 2020). We augment the dataset with five additional properties using RDKit (Landrum et al., 2023), including the drug-likeness score QED (Bickerton et al., 2012; Wildman & Crippen, 1999) and topological polar surface area (TPSA) and refer to the resulting $M = 6$ dataset as Caco-2+. In many cases, subsets of these properties (e.g., permeability and TPSA) will be inversely correlated and thus compete with one another during optimization. In late-state drug optimization, the trade-offs become more dramatic and as more properties are added (Sun et al., 2022). Demonstrating effective sampling of Pareto-optimal solutions in this setting is thus of great value.

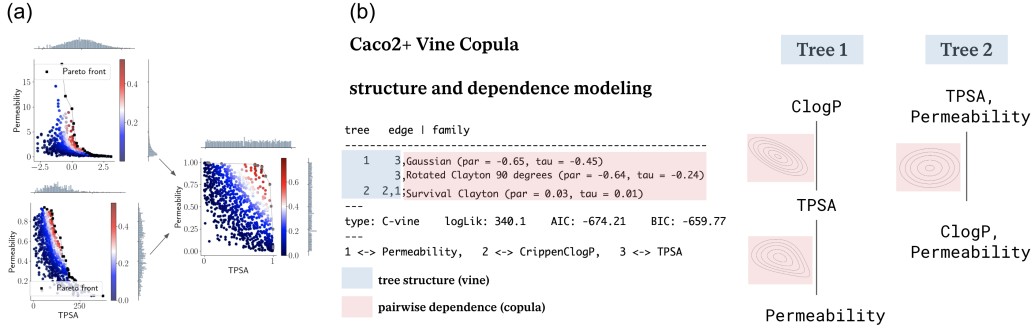

Figure 3: (a). Regardless of the distributions of the marginals, the CDF score from a copula is the same. (b) An example of explicitly encoding domain knowledge in a BO procedure by imposing the blue tree structure (specifying the matrix representation of the vine) and pink selection of pairwise dependencies (choice of parametric/non-parametric family).

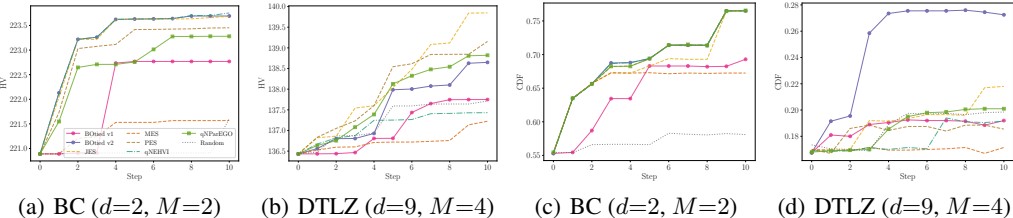

(a) BC ($d$=2, $M$=2)   (b) DTLZ ($d$=9, $M$=4)   (c) BC ($d$=2, $M$=2)   (d) DTLZ ($d$=9, $M$=4)

Figure 4: (a, b): HV metric vs. iterations. (c, d): CDF metric vs. iterations. Values shown are averages over five random seeds.

## 4.1 COPULAS IN BO

In the low-data regime, empirical Pareto frontiers tend to be noisy. When we have access to domain knowledge about the objectives, we can use it to construct a model-based Pareto frontier using vine copulas. This section describes how to incorporate (1) the known correlations among the objectives to specify the tree structure (vine) and (2) the pairwise joint distributions (including the tail behavior), approximately estimated from domain knowledge, when specifying the copula models.

The advantages of integrating copula-based estimators for our metric and acquisition function are threefold: (i) scalability from the convenient pair copula construction of vines, (ii) robustness wrt marginal scales and transformations thanks to inherent copula properties 3.2 and Eq. 2, and (iii) domain-aware copula structures from the explicit encoding of dependencies in the vine copula matrix, including choice of dependence type (e.g., low or high tail dependence).

Fig. 3 illustrates the use of copulas in the context of optimizing multiple objectives in drug discovery, where data tends to be sparse. In panel (a) we see that, thanks to the separate estimation of marginals and dependence structure, different marginal distributions have the same Pareto front in the PIT space, in which we evaluate our CDF scores. Hence, with copula-based estimators, we can guarantee robustness without any overhead for scalarization or standardization of the data as required by other counterparts. In panel (b) we show how we can encode domain knowledge of the interplay between different molecular properties in the Caco2+ dataset. Namely, permeability is often highly correlated with ClogP and TPSA, with positive and negative correlation, respectively, which is even more notable at the tails of the data (see panel (a) and Appendix F). Such dependence can be encoded in the vine copula structure and in the choice of copula family for each pair. For example, we specified a rotated Clayton copula so that the tail dependence between TPSA and permeability is preserved.

**New flexible test function**   We design a dataset named *CopulaBC* to explore an ideal case for BOTIED ranking, in which we do not incur error from specifying an incorrect CDF model. The objectives follow a known joint distribution, recoverable using the true data-generating model for the marginals and for the copula. For particular copula families, this dataset also enables analyses of the dependency structure of the objectives out to the tails. We set $d = 2$, $M = 2$ for simplicity but a higher dimensional dataset can be generated with an analogous approach. See Appendix B for details.

## 4.2 RESULTS AND DISCUSSION

We compare the performance of BOTIED with baselines in terms of both the HV and the CDF indicators Table 1 and Fig. 4. Although there is no single best method across all the datasets, the best numbers are consistently achieved by either BOTIED v1 or v2 with NParEGO being a close competitor. The NEHVI performance visibly degrades as $M$ increases and it becomes increasingly slow. From the entropy baselines only JES has competitive CDF score but only in the BC data. As with EHVI, performance degrades as M increases. In addition to being on par with commonly used acquisition functions, BOTIED is significantly faster than NEHVI and JES (Fig. 5).

There are two main benefits to using the CDF metric rather than HV for evaluation. First, the CDF is bounded between 0 and 1, with scores close to 1 corresponding to the discovered solutions closest to

Table 1: HV indicators (computed in the original units) and CDF indicators across synthetic datasets. Higher is better and best per column is marked in bold. We report the average metric across five random seeds along with their standard deviations in parentheses.

| | BC (M=2) | | DTLZ (M=4) | | DTLZ (M=6) | | DTLZ (M=8) | |
|---|---|---|---|---|---|---|---|---|
| | CDF | HV | CDF | HV | CDF | HV | CDF | HV |
| **BOTIED v1** | **0.76 (0.06)** | 1164.43 (174.37) | 0.2 (0.1) | 0.42 (0.03) | **0.33 (0.09)** | 0.52 (0.02) | **0.2 (0.08)** | 0.93 (0.02) |
| **BOTIED v2** | 0.74 (0.08) | **1205.3 (120.46)** | **0.24 (0.2)** | **0.45 (0.05)** | 0.32 (0.08) | 0.58 (0.03) | 0.19 (0.1) | 0.91 (0.03) |
| **NParEGO** | 0.73 (0.09) | 993.31 (178.16) | 0.20 (0.07) | 0.4 (0.03) | 0.29 (0.03) | **0.69 (0.02)** | 0.13 (0.07) | **1.05 (0.02)** |
| **NEHVI** | 0.73 (0.07) | 1196.37 (98.72) | 0.21 (0.02) | 0.44 (0.04) | – | – | – | – |
| **Random** | 0.71 (0.11) | 1204.99 (69.34) | 0.1 (0.05) | 0.22 (0.03) | 0.10 (0.03) | 0.55 (0.02) | 0.13 (0.07) | 0.96 (0.02) |
| | Caco2+ (M=3) | | Penicillin (M=3) | | CopulaBC (M=2) | | | |
| | CDF | HV | CDF | HV | CDF | HV | | |
| **BOTIED v1** | 0.58 (0.06) | 11645.63 (629.0) | 0.48 (0.02) | 319688.6 (17906.2) | 0.9 (0.03) | 1.08 (0.03) | | |
| **BOTIED v2** | **0.60 (0.06)** | 11208.57(882.21) | **0.49 (0.02)** | 318687.7 (17906.2) | **0.9 (0.01)** | 1.09 (0.02) | | |
| **NParEGO** | 0.56 (0.05) | 12716.2 (670.12) | 0.28 (0.09) | **332203.6 (15701.52)** | 0.87 (0.01) | 1.1 (0.01) | | |
| **NEHVI** | 0.54 (0.06) | **13224.7 (274.6)** | 0.24 (0.05) | 318748.9 (2868.64) | 0.88 (0.02) | 1.1 (0.01) | | |
| **Random** | 0.57 (0.07) | 11425.6 (882.4) | 0.32 (0.02) | 327327.9 (17036) | 0.88 (0.02) | 1.08 (0.01) | | |

our approximate Pareto front.[3] Unlike with HV, for which the scales do not carry information about the internal ordering, the CDF values have an interpretable scale. Second, applying the CDF metric for different tasks (datasets), we can easily assess how the acquisition performance varies with the specifics of the data, assuming the GP and copula have been properly fit.

We stress-test BOTIED in a series of ablation studies in Appendix G. In particular, we vary the number of MC posterior samples and find that BOTIED v1 is robust to the number of posterior samples, i.e., the multivariate ranks associated with the best-fit copula model do not change significantly with varying numbers of samples. When the posterior shape is complex such that many MC samples are required to fully characterize the posterior, BOTIED v2 (in which the copula is fit on the mean of the posterior samples) is more appropriate than v1.

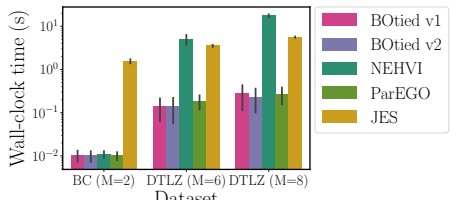

Figure 5: Wall-clock time per single call of acquisition function. Error bars are standard deviations across five repeated calls.

**Limitations** There is a trade-off between flexibility and complexity when fitting the copula model. As the number of objectives grow, so does the number of modeling choices that need to be made (pair copulas, parameters etc). For efficiency, in experiments we always pre-select the copula family (Gaussian or KDE), which reduces time complexity without impacting BOTIED's performance.

## 5 CONCLUSION

We introduce a new perspective on MOBO based on the multivariate CDF. We view the role of the multi-objective acquisition function as producing multivariate ranks in the presence of noise (surrogate uncertainties). To obtain the ranks, we fit a multivariate CDF on the surrogate predictions and extract the ranks associated with the CDF scores. The CDF can be fit efficiently using copulas. We propose a new Pareto-compliant CDF indicator as well as a CDF-based multi-objective acquisition function. We have demonstrated that our CDF-based estimation of the non-dominated regions allows for greater flexibility, robustness, and scalability compared to existing acquisition functions.

This method is general and lends itself to a number of immediate extensions. First, we can encode dependencies between objectives, estimated from domain knowledge, into the graphical vine model. Second, we can accommodate discrete-valued objectives, e.g., by introducing continuous relaxations of the discrete objectives. Finally, as many applications carry noise in the input as well as the function of interest, accounting for input noise through the established connection between copulas and multivariate value-at-risk (MVaR) estimation will be of great practical interest. Finally, whereas we have focused on gradient-free evaluations of our BOTIED score, the computation of our acquisition function is differentiable for many parametric copula families and admits gradient-based optimization over the input space.

---

[3]Binois et al. (2015) shows that the zero level lines $F(\cdot) = 0$ correspond to the estimated Pareto front in a minimization setting, which are equivalent to the one level lines $F(\cdot) = 1$ in the maximization setting.

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

# A  PROPERTIES OF THE CDF INDICATOR

## A.1  THEOREM 1: PARETO COMPLIANCE OF THE CDF INDICATOR

We state Theorem 1 again and provide the proof here.

**Theorem 1**: For any arbitrary approximation sets $A \in \mathcal{X}$ and $B \in \mathcal{X}$ where $\mathcal{X} \subset \mathbb{R}^d$, the following holds:
$$A \succcurlyeq B \wedge B \nsucceq A \Rightarrow I_F(A) \geq I_F(B).$$

*Proof.* If we have $A \succcurlyeq B \wedge B \nsucceq A$, then the following two conditions hold: $\forall \boldsymbol{x}' \in B \,\exists \boldsymbol{x} \in A : \boldsymbol{x} \succcurlyeq \boldsymbol{x}'$ and $\exists \mathbf{x} \in A$ s.t. $\nexists \boldsymbol{x}' \in B : \boldsymbol{x}' \succcurlyeq \mathbf{x}$. Recall that the weak Pareto dominance $\mathbf{x} \succcurlyeq \mathbf{x}'$ implies that $\forall i \in [M] :$ $f_i(\boldsymbol{x}) \geq f_i(\boldsymbol{x}')$. From the definition and fundamental property of the CDF being a monotonic non-decreasing function, it follows that $\forall i \in [M] : f_i(\boldsymbol{x}) \geq f_i(\boldsymbol{x}') \Rightarrow F_{\mathbf{Y}}(\boldsymbol{x}) \geq F_{\mathbf{Y}}(\boldsymbol{x}')$.

Define the set of non-dominated solutions in $B$, $\mathcal{P}_B := \{\boldsymbol{x} \in B, \forall \boldsymbol{x}' \in B : \boldsymbol{x} \succeq \boldsymbol{x}'\}$. Note that $I_F(B) = I_F(\mathcal{P}_B) = I_F(\{\boldsymbol{z}\})$ for any $\boldsymbol{z} \in \mathcal{P}_B$. Now let $\boldsymbol{x}_B \in \mathcal{P}_B$. There is $\boldsymbol{x}_A \in A$ such that $\boldsymbol{x}_A \succeq \boldsymbol{x}_B$, and we have that $F_{\mathbf{Y}}(\boldsymbol{x}_A) \geq F_{\mathbf{Y}}(\boldsymbol{x}_B)$. By definition, $I_F(A) \geq I_F(\{\boldsymbol{x}_A\})$ so we have $I_F(A) \geq I_F(\{\boldsymbol{x}_A\}) \geq I_F(\{\boldsymbol{x}_B\}) = I_F(B)$ as desired. $\square$

## A.2  COROLLARY 2: INVARIANCE UNDER MONOTONIC TRANSFORMATIONS

This proof closely follows the one in (Haugh, 2016).

**Corollary 2:** Let $Y_1, Y_2$ be continuous random variables with copula $C_{Y_1, Y_2}$. If $\alpha, \beta : \mathbb{R} \to \mathbb{R}$ are strictly increasing functions, then:
$$C_{\alpha(Y_1), \beta(Y_2)} = C_{Y_1, Y_2} \tag{A.1}$$
where $C_{\alpha(Y_1), \beta(Y_2)}$ is the copula function corresponding to variables $\alpha(Y_1)$ and $\beta(Y_2)$.

*Proof.* We first note that for the distribution function of $\alpha(Y_1)$ it holds that
$$F_{\alpha(Y_1)} = P(\alpha(Y_1) \leq y_1) = P(Y_1 \leq \alpha^{-1}(y_1)) = F_{Y_1}(\alpha^{-1}(y_1)) \tag{A.2}$$
and analogously,
$$F_{\beta}(Y_1)(y_1) = F_{Y_1}(\beta^{-1}(y_1)) \tag{A.3}$$

From Sklar's theorem, we have that for all $y_1, y_2 \in \mathbb{R}$

$$\begin{aligned}
C_{\alpha(Y_1)\beta(Y_2)}(F_{\alpha(Y_1)}(y_1), F_{\beta(Y_2)}(y_2)) &= F_{\alpha(Y_1)\beta(Y_2)}(y_1, y_2) \\
&= P(\alpha(Y_1) \leq y_1, \beta(Y_2) \leq y_2) \\
&= P(Y_1 \leq \alpha^{-1}(y_1), Y_2 \leq \beta^{-1}(y_2)) \\
&= F_{Y_1, Y_2}(\alpha^{-1}(y_1), \beta^{-1}(y_2)) \\
&= C_{Y_1, Y_2}(F_{Y_1}(\alpha^{-1}(y_1)), F_{Y_2}(\beta^{-1}(y_2))) \\
&= C_{Y_1, Y_2}(F_{\alpha(Y_1)}(y_1), F_{\beta(Y_2)}(y_2))
\end{aligned}$$

Equalities one and five follow from Sklar's theorem. In the third equality we make use of fact that $\alpha$ and $\beta$ are increasing functions. The last equality follows from Eq. A.2 and Eq. A.2. $\square$

# B  COPULABC TEST FUNCTION

We design and release a dataset named CopulaBC for MOBO benchmarking. CopulaBC explores an ideal case for BOTIED ranking, in which we do not incur error from specifying an incorrect CDF model. The objectives follow a known joint distribution. The generation process is as follows:
$$U \sim C \tag{B.1}$$
$$\forall i \in [M] : Y_i = F_i^{-1}(U_i) \tag{B.2}$$
$$X = g^{-1}(Y), \tag{B.3}$$

where $C$ is the assigned copula model and $g$ is the function mapping inputs in $\mathcal{X} \subset \mathbb{R}^d$ to outputs in $\mathbb{R}^M$. In words, we first sample the uniform marginals in $\mathbb{R}^M$ from a copula model of choice in Eq. B.1. We may impose the correlation structure of interest in the copula model; for instance, we may assign a flipped

Clayton copula if we want to emulate a heavy right tail behavior, as is common in biophysical properties (Jain et al., 2017). The uniform marginals are then converted into the observation space via the marginal quantile functions $F_i^{-1}$ for each dimension $i \in [M]$. Lastly, the resulting $Y \in \mathbb{R}^M$ are mapped back to the input $X \in \mathcal{X} \subset \mathbb{R}^d$ via the inverse model $g^{-1}$.

For CopulaBC, we use the flipped Clayton copula with one parameter for $C$. Choosing within the Archimedean copula family offers the advantage of analytic expressions of the Pareto front, as their level lines admit an analytic form. We specifically opt for the Branin-Currin function for $g^{-1}$ (hence the "BC" in the name CopulaBC), but other functions with bigger $d$ and $M$ may be used as long as the support of $F_i$ is consistent with the image of $g$ (i.e., the space of $Y$ is matched between $F_i$ and $g$). Because we restrict the domain of Branin-Currin (the image of our $g$) to $[0, 1]^d$, we chose $F_i(\cdot) = \text{Beta}(\cdot; \alpha = 2, \beta = 2)$ for all $i \in [M]$. We can also choose from a number of parametric families (e.g., exponential, Gaussian, Student's t) because, thanks to the nature of copulas, the marginals have no impact on the dependence structure.

Our copula dataset ensures that the objective $Y$ follows a distribution that can be fit easily using the true data-generating model for the marginals and for the copula (e.g., beta and Clayton, respectively, for CopulaBC). However, this copula-based test function is general and allows for "mimicking" a variety of scenarios in stress tests or ablation studies, depending on the application. A practitioner can control for the following:

1. choice of dependence structure, high or low tail dependence (very common in finance for example)

2. choice of distribution $F_i$ for each of the marginals (for example, drug-related properties often follow zero-inflated distributions with upper tails)

3. choice of the vine structure; as illustrated in Fig. 3, we can decide on the pairwise factorization and enforce dependencies between properties via the vine copula matrix.

## C (VINE) COPULA OVERVIEW AND EXAMPLE

According to Sklar's theorem Sklar (1959), the joint density of any bivariate random vector $(X_1, X_2)$, can be expressed as

$$f(x_1, x_2) = f_1(x_1)f_2(x_2)c\left(F_1(x_1), F_2(x_2)\right) \tag{C.1}$$

where $f_i$[4] are the marginal densities, $F_i$ the marginal distributions, and $c$ the copula density.

That is, any bivariate density is uniquely described by the product of its marginal densities and a *copula density*, which is interpreted as the *dependence structure*. For self-containment of the manuscript, we borrow an example from Tagasovska et al. (2023). Fig. C.7 illustrates all of the components representing the joint density. As a benefit of such factorization, by taking the logarithm on both sides, one can estimate the joint density in two steps, first for the marginal distributions, and then for the copula. Hence, copulas provide a means to flexibly specify the marginal and joint distribution of variables. For further details, please refer to Aas et al. (2009); Joe et al. (2010).

There exist many parametric representations through different copula families, however, to leverage even more flexibility, in this paper, we focus on the kernel-based nonparametric copulas of Geenens et al. (2017). Eq. C.1 can be generalized and holds for any number of variables. To be able to fit densities of more than two variables, we make use of the pair copula constructions, namely *vines*; hierarchical models, constructed from cascades of bivariate copula blocks Nagler et al. (2017). According to Joe (1997); Bedford & Cooke (2002a), any $M$-dimensional copula density can be decomposed into a product of $\frac{M(M-1)}{2}$ bivariate (conditional) copula densities. Although such factorization may not be unique, it can be organized in a graphical model, as a

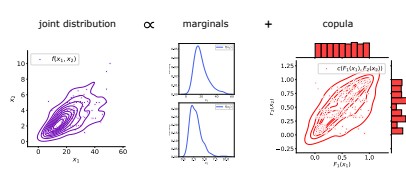

Figure 6: Expressing joint densities with copulas.

sequence of $M - 1$ nested trees, called *vines*. We denote a tree as $T_m = (V_m, E_m)$ with $V_m$ and $E_m$ the sets of nodes and edges of tree $m$ for $m = 1, \ldots, M - 1$. Each edge $e$ is associated with a bivariate copula. An example of a vine copula decomposition is given in Fig. 7.

In practice, in order to construct a vine, one chooses two components:

1. the structure, the set of trees $T_m = (V_m, E_m)$ for $m \in [M - 1]$

2. the pair-copulas, the models for $c_{j_e, k_e | D_e}$ for $e \in E_m$ and $m \in [M - 1]$.

---

[4]In this section, we use the standard notations for densities ($f$) and distributions ($F$) as commonly done in the copula literature.

Corresponding algorithms exist for both of those steps and in the rest of the paper, we assume consistency of the vine copula estimators for which we use the implementation by Nagler & Czado (2016), namely its Python version -pyvinecopulib.

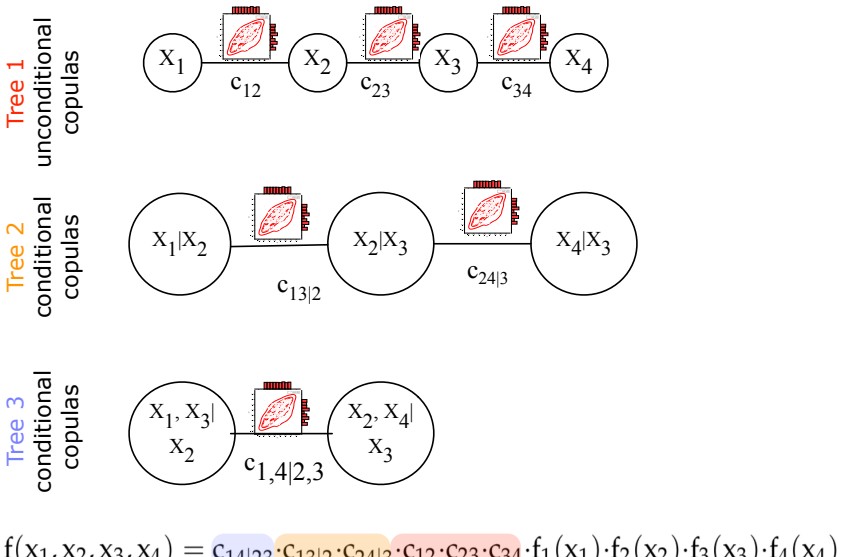

$$f(x_1, x_2, x_3, x_4) = c_{14|23} \cdot c_{13|2} \cdot c_{24|3} \cdot c_{12} \cdot c_{23} \cdot c_{34} \cdot f_1(x_1) \cdot f_2(x_2) \cdot f_3(x_3) \cdot f_4(x_4)$$

Figure 7: Multivariate joint density factorized with a vine copula.

### C.1 COMPLEXITY OF THE COPULA ESTIMATION

The complexity for fitting the vine copulas as currently implemented scales as $O(\text{number of points} M \text{ vine depth})$ in the case of density estimation. Both estimation and sampling involve a double loop over $M$ and vine depth with an internal step scaling linearly with number of points. As the reviewer appropriately pointed out, the computational complexity is highly impacted by $L$ (number of predictive samples). For BOtied v1, this translates to $O(nLM \text{ vine depth})$, where $n$ is the number of query candidates, while for BOtied v2, we use the expectation of the posterior samples only, so the complexity remains as $O(n \times M \times \text{vine depth})$. We have evaluated the impact of varying $L$ and the $q$ batch size in **??**.

## D RELATED WORK

Table 2: Comparison of BOtied with related work

| Type of groundwork | Scoring method | MO criteria | Scalability with M | Scale invariance | Bayesian optimization | Non GP surrogates |
|---|---|---|---|---|---|---|
| Multivariate ranks/CDF, Copula, Copula space | BOtied (this work) | ✓ | ✓ | ✓ | ✓ | ✓ |
| Copula space, Game theory | Kalai-Smorodinsky MO (Binois et al., 2020) | ✓ | ✓ | ✓ | ✓ | ✗ |
| Multivariate ranks | Aggregate Rank (Kukkonen & Lampinen, 2007) | ✓ | ✓ | ✓ | ✗ | ✗ |
| | Ordinal BO (Picheny et al., 2019) | ✓ | ✗ | ✗ | ✓ | ✓ |
| Information Theoretic | Joint Entropy Search (Tu et al., 2022; Hvarfner et al., 2022) | ✓ | ✗ | ✓ | ✓ | ✗ |
| | Predictive Entropy Search (Hernández-Lobato et al., 2016a) | ✓ | ✗ | ✗ | ✓ | ✗ |
| | Max-Value Entropy Search (Belakaria et al., 2019) | ✓ | ✗ | ✗ | ✓ | ✗ |
| Hypervolume | EHVI variants (Daulton et al., 2021; 2022b) | ✓ | ✗ | ✗ | ✓ | ✓ |
| Random scalarization | ParEGO (Knowles, 2006) | ✓ | ✓ | ✗ | ✓ | ✓ |
| Boundary distance | SVM-variants (Miranda & Von Zuben, 2016; Shilton et al., 2018) | ✓ | ✓ | ✗ | ✓ | ✗ |
| Maxmin, Pareto Indicator | Pareto improvement , EmaX (Bautista, 2009) | ✓ | ✓ | ✗ | ✓ | ✓ |
| | Maximin improvement (Svenson, 2011) | ✓ | ✓ | ✗ | ✓ | ✓ |
| Completeness | Averaged completeness indicator (Svenson, 2011) | ✓ | ✗ | ✓ | ✓ | ✗ |
| | Estimated completeness indicator improvement (Svenson, 2011) | ✓ | ✓ | ✓ | ✓ | ✓ |

# E  ALGORITHM

---

**Algorithm 1:** MOBO with BOTIED: a CDF-based acquisition function

---

1: **Input:** Probabilistic surrogate $\hat{f}$, initial data $\mathcal{D}_0 = \{(\boldsymbol{x}_n, \boldsymbol{y}_n)\}_{n=1}^{N_0}$, $\mathcal{X} \subset \mathbb{R}^d$, $\mathcal{Y} \subset \mathbb{R}^M$

2: **Output:** Optimal selected subset $\mathcal{D}_T$.

3: Fit the initial surrogate model $\hat{f}(\boldsymbol{x}_i)$ on $\mathcal{D}_0$.

4: **for** $\{t = 1, \ldots, T\}$ **do**

5:      Sample the candidate pool $\boldsymbol{x}_1, \cdots, \boldsymbol{x}_N \in \mathcal{X}$

6:      **for** $\{i = 1, \ldots, N\}$ **do**

7:          Evaluate $\hat{f}$ on the candidate pool to obtain the posterior $p(f(\boldsymbol{x}_i)|\mathcal{D}_{t-1})$

8:          Draw $L$ predictive samples $\hat{f}_i^{(j)} \sim p(f(\boldsymbol{x}_i)|\mathcal{D}_{t-1})$, for $j \in [L]$

9:      **end for**

10:      Obtain uniform marginals $\{u_i^{(j)}\}_{i \in [N], j \in [L]}$ from the pooled samples $\{\hat{f}_i^{(j)}\}_{i \in [N], j \in [L]}$

11:      Version 1: Fit a vine copula $\hat{C}$ on the uniform marginals on the sample level, $\{u_i^{(j)}\}_{i \in [N], j \in [L]}$.

         Version 2: Fit a vine copula $\hat{C}$ on the mean-aggregated uniform marginals, $\{\frac{1}{L} \sum_{j=1}^{L} u_i^{(j)}\}_{i \in [N]}$.

12:      **for** $\{i = 1, \ldots, N\}$ **do**

13:          Version 1: Compute the expected CDF score $\mathcal{S}(\boldsymbol{x}_i) = \frac{1}{L} \sum_{j=1}^{L} \hat{C}\left(u_i^{(j)}\right)$

         Version 2: Compute the CDF score of the mean ranks $\mathcal{S}(\boldsymbol{x}_i) = \hat{C}\left(\frac{1}{L} \sum_{j=1}^{L} u_i^{(j)}\right)$

14:      **end for**

15:      $i^\star \leftarrow \arg\max_{i \in [N]} \mathcal{S}(\boldsymbol{x}_i)$

16:      $\mathcal{D}_t \leftarrow \mathcal{D}_{t-1} \cup \{(\boldsymbol{x}_{i^\star}, \boldsymbol{y}_{i^\star})\}$

17: **end for**

18: **return** $\mathcal{D}_T$

---

# F  EXPERIMENTAL DETAIL

We executed batched BO simulations with a batch size of $B = 4$ for all the experiments. The number of iterations $T$ varied across the experiments. Other parameters include: the initial data size $N_0$, the size of the pool $N$, and the number of predictive posterior samples $L$. We fixed the size of the pool relative to the selected batch, at $N/B = 100$. We also fixed $L = 20$, which was found to yield good sample coverage and a stable BOTIED acquisition value.

Unless otherwise stated, the surrogate model was a multi-task Gaussian process (MTGP) with a Matern kernel implemented in `BoTorch` (Balandat et al., 2020) and `GPyTorch` (Gardner et al., 2014). The inputs and outputs were both scaled to the unit cube for fitting the MTGP, but the outputs were scaled back to their natural units for evaluating the respective acquisition functions.

## F.1  BRANIN-CURRIN

Branin-Currin ($d = 2$, $M = 2$; (Belakaria et al., 2019)) is a composition of the Branin and Currin functions featuring a concave Pareto front (in the maximization setting). We maximize

$$f_1(x_1, x_2) = -\left(x_2 - \frac{5.1}{4\pi^2}x_1^2 + \frac{5}{\pi}x_1 - r\right)^2 + 10(1 - \frac{1}{8\pi})\cos(x_1) + 10$$

$$f_2(x_1, x_2) = -[1 - \exp\left(-\frac{1}{2x_2}\right)]\frac{2300x_1^3 + 1900x_1^2 + 2092x_1 + 60}{100x_1^3 + 500x_1^2 + 4x_1 + 20},$$

where $x_1, x_2 \in [0, 1]$. We used $T = 30$.

## F.2  DTLZ

For the DTLZ problem, we took DTLZ2 ($d = 9$, $M \in \{4, 6, 8\}$; (Deb & Gupta, 2005)) and used $T = 20$.

### F.3 PENICILLIN PRODUCTION

For the Penicillin production problem ($d = 7$, $M = 3$; (Liang & Lai, 2021)), we used $T = 10$.

### F.4 CACO2+

For the Caco2 problem ($M = 3$; Wang et al., 2016a), we use $T = 10$. The objective is to identify molecules with maximum cell permeability. Here, permeability describes the degree to which a molecule passes through a cellular membrane. This property is critical for drug discovery (DD) programs where the disease protein being targeted resides within the cell (intracellularly). In each experiment, a molecule $x_i$ is applied to a monolayer of Caco2 cells and, after incubation, the concentration $c$ of $x_i$ is measured on both the input and output side of the monolayer, giving $c_{\text{in}}$ and $c_{\text{out}}$(Van Breemen & Li, 2005). The ratio $c_{\text{out}}/c_{\text{in}}$ is then treated as the final permeability label $y_i^p$.

Cellular membranes are composed of a complex mixture of lipids and other biomolecules. In order to enter and (passively) diffuse through a membrane, molecule $x_i$ should interact favorably with these biomolecules and/or avoid disrupting their packing structure. Increasing the lipophilicity (logP) of $x_i$ is thus one strategy to increase permeability. However, increasing logP often results in promiscuous binding of $x_i$ to non-disease related proteins, which can lead to undesired side-effects. As such, we seek to minimize the computed logP (clogP, $y_i^l$) in our optimization task and note that this could directly compete with (i.e., harm) permeability.

Lastly and related, common objectives during MPO in DD settings include increasing the affinity and specificity of target binding. As opposed to non-specific lipophilic interactions as above, polar contacts (such as hydrogen bonds) between drug molecules and proteins often result in higher affinity and more specific binding. We compute the topological polar surface area (TPSA, $y_i^t$) of each candidate $x_i$ as one indicator of its ability to form such interactions and seek to maximize it in our optimization. As with decreasing logP, increasing TPSA can negatively impact permeability and we thus consider it a competing objective.

It is important to note that the treatment of each of these optimization tasks as unidirectional (max or min) is a simplification of many practical DD settings. There is often an acceptable range of each value that is targeted, and leaving the bounds in either direction can be problematic for complex reasons. We direct the reader to D. Segall (2012) for a comprehensive review.

For fitting the MTGP on the Caco2+ data, we represent each input molecule as a concatenation of fingerprint and fragment feature vectors, known as fragprints (Thawani et al., 2020) and use the Tanimoto kernel implemented in GAUCHE (Griffiths et al., 2022).

### F.5 DETAILS ON WALL CLOCK TIME

**Details for Fig. 5**. For all acquisition functions, we report the wall clock time per single acquisition function evaluation as computed on a Tesla V100 SXM2 GPU (16GB RAM) and an Intel Xeon CPU @ 2.30GHz (240GB RAM). A single call takes in the surrogate inference results for the candidate pool as well as the previously evaluated points and computes the acquisition scores.

- BC M=2: q batch size = 4, number of predictive samples=40, initial n = 10, pool size = 40

- DTLZ M=4: q batch size = 4, number of predictive samples=20, initial n = 50, pool size = 40

- DTLZ M=6: q batch size = 4, number of predictive samples=20, initial n = 50, pool size = 40.

Therefore, a single call for BOtied includes the copula computation and this is reflected in Figure 4. The complexity for fitting the copula scales as $O(\text{numbpoints} M \text{vinedepth})$, where numpoints is $nL$ for BOtied v1 and $n$ for BOtied v2. The copula is refit at every iteration to take advantage of the additional data. In the case of a batch selection of q candidates within an iteration, the computation for BOtied increases linearly with every iteration, since this adds q points to the copula at each iteration. The L predictive samples of those q candidates are also included in the copula fit.

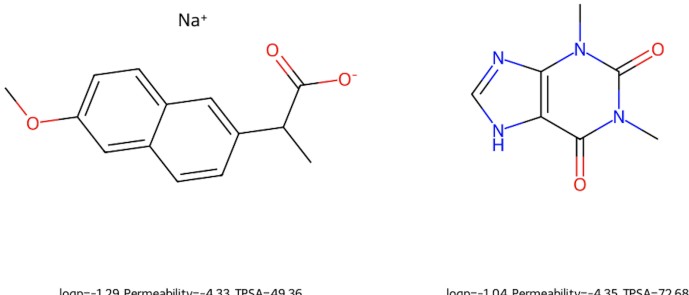

logp=-1.29 Permeability=-4.33 TPSA=49.36          logp=-1.04 Permeability=-4.35 TPSA=72.68

Figure 8: Examples of molecules with "good"/desirable TPSA, permeability, and ClogP values.

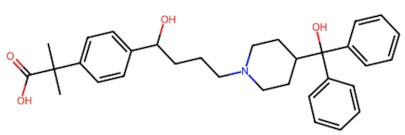

logp=5.51 Permeability=-6.51 TPSA=81.00

Figure 9: An example of a molecule with "bad" TPSA, permeability, and ClogP values.

## G    ABLATION STUDIES

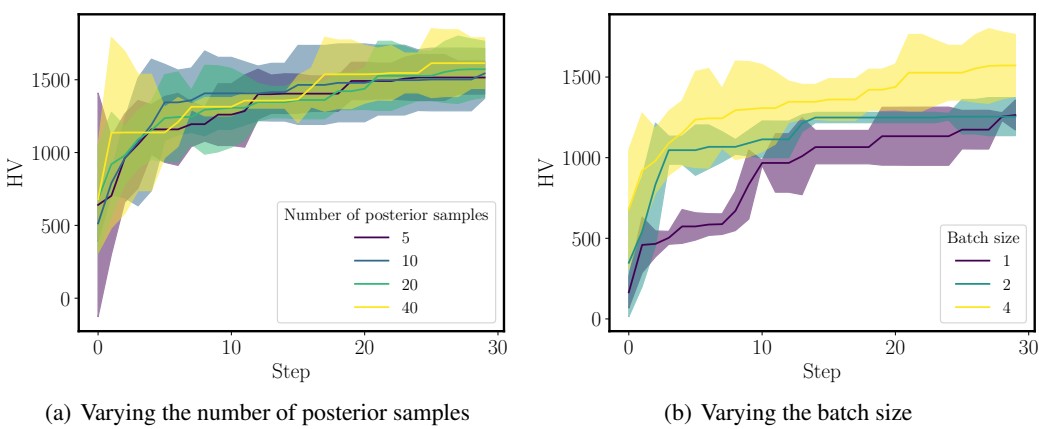

(a) Varying the number of posterior samples          (b) Varying the batch size

Figure 10: Ablation studies for BOtied v1. (a) BOtied is robust to the number of posterior samples drawn. (b) Increasing the batch size improves acquisition, particularly as it improves the CDF fit quality in earlier iterations.

