# OpenReview forum: "BOtied: Multi-objective Bayesian optimization with tied multivariate ranks"
_ICLR.cc/2024/Conference — Submitted to ICLR 2024_

### Official Review · Reviewer_3FiX · 2023-10-26

**Soundness:** 3 good
**Presentation:** 3 good
**Contribution:** 3 good
**Rating:** 6
**Confidence:** 5

**Summary:**

The authors consider concurrent optimization, with multiple objectives, within a Bayesian optimization framework. Their primary emphasis lies on introducing a novel acquisition function rooted in multivariate ranks. This innovative approach aims to alleviate the computational burden tied to the hypervolume or entropy calculations in established criteria. The process entails sampling from the posterior distribution to estimate uniform marginals, followed by rank assignment post a copula transformation. The study includes a comprehensive evaluation across various simplified scenarios, along with a selection of more complex test cases.

**Strengths:**

- Detailed description of the method.
- Summary of relationships with the state of the art

**Weaknesses:**

- Mostly toy examples are provided, e.g., DTLZ test function with a simple Pareto front.
- Only discrete inputs are considered

**Questions:**

Figure 3a is too small.

The method only seems to work on discrete sets, according to Algorithm 1. But some problems are continuous, like Branin-Currin or the DTLZ test problems. The adaptation is not clearly discussed.

It is preferable to show progress curves over iterations rather than fixed snapshots.

Typos
P7: being the whether the

---

### Official Review · Reviewer_x952 · 2023-11-03

**Soundness:** 2 fair
**Presentation:** 2 fair
**Contribution:** 2 fair
**Rating:** 5
**Confidence:** 3

**Summary:**

The paper proposes a CDF based acquisition function for black-box multi-objective optimization (MOO). The basic idea is to use the CDF function as a criterion for the next point selection and the authors discuss a connection between CDF and multivariate rank. To evaluate the CDF, a copula based approach is introduced by which the authors claim scalability and invariance properties are obtained.

**Strengths:**

Experimental results seemingly show a good result.

Introducing copula-based computations into multi-objective BO is seemingly novel.

**Weaknesses:**

Overall, the description is unclear. For example, although the CDF plays a key role, the distribution of CDF is not explicitly written in the methodology section. In appendix, I found the authors used multi-task GP. Another example is \hat{f} in (3.8) suddenly appear without explanation (explained in the experimental section), by which the definition of the acquisition function becomes unclear at that point.

As mentioned above, the authors used multi-task GP, but for GP based MOO (i.e., BO), except for a few studies (e.g., Shah and Ghahraman 2016), typically uses independent GPs for multiple objectives. When the independent GPs are used, CDF is quite easy to evaluate (just by multiplying one dimensional CDFs). However, the author does not mention importance of modeling correlation among objectives. In practice, independent GPs often show sufficient (or better) performance compared with multi-task GP (for which task-correlation must be carefully tuned to achieve good performance).

Rationale behind the CDF based criterion is unclear. For simplicity, consider the case of independent Gaussian for each objective, which would be the simplest special case. Then, CDF becomes just a multiplication of each dimension of CDF, which intensely seeks a `specific direction' of the output space though, in MOO, the Pareto frontier should exist in a variety of direction of the output space. In this sense, in my current understanding, the proposed acquisition function is not appropriate for exploring the entire Pareto frontier that can be widely distributed in the output space. Even when correlated model is used, this problem would not be avoided.

**Questions:**

In the experiment section, the authors mention the predictive mean is used for CDF calculation. Does that mean \hat{f} is set as the posterior mean of GPs?

---

### Official Review · Reviewer_vKP1 · 2023-11-05

**Soundness:** 1 poor
**Presentation:** 2 fair
**Contribution:** 2 fair
**Rating:** 3
**Confidence:** 4

**Summary:**

In multi-objective Bayesian optimization, existing acquisition functions scale poorly to a large number of objectives. To address this, the paper introduces the CDF indicator as a Pareto-compliant performance criterion to measure the quality of Pareto sets and proposes an acquisition function called BOTIED, which can be implemented efficiently using copulas.

**Strengths:**

1. This paper focuses on improving the computational efficiency of the acquisition function, which is an important issue in multi-objective optimization.
2.The idea of incorporating domain knowledge and utilizing dependency structures of objectives in optimization seems new.

**Weaknesses:**

1. Experimental results do not fully demonstrate the advantage of BOTIED. As for optimization results, in table I, in DTLZ(M=6) and DTLZ(M=8), BOTIED underperforms NParEGO on HV. BOTIED even achieves lower HV than Random in DTLZ(M=8). The authors claim that BOTIED is computational efficient. However, according to Figure 5, ParEGO requires similar computational time to BOTIED.
2. How to estimate high-dimensional CDFs with copulas is critical in BOTIED. However, there is a lack of description on this issue in the paper.

**Questions:**

1. Why is the HV and CDF data inconsistent in Figure 4 and Table 1 in BC(M=2) and DTLZ(M=4)?
2. The authors mention that the CDF indicator does not discriminate sets with the same best element. This seems questionable because the purpose of multi-objective optimization is to identify a solution set instead of a single best element.
3. The authors argue that ‘the CDF indicator is invariant to arbitrary monotonic transformations of the objectives, whereas the HV indicator is highly sensitive to them.’ However, I believe it is the HV indicator for a set instead of the HV for a single point that matters in multi-objective optimization. What is the problem of the HV indicator being sensitive to monotonic transformations of the objectives?

---

### Official Review · Reviewer_8cpR · 2023-11-09

**Soundness:** 3 good
**Presentation:** 2 fair
**Contribution:** 2 fair
**Rating:** 3
**Confidence:** 4

**Summary:**

This paper solves a multi-objective Bayesian optimization problem, which attempts to find a Pareto frontier on a metric space.  Since identifying a Pareto frontier and calculating a hypervolume are time-consuming, this research proposes a novel method using the highest multivariate rank, which is the outermost level line of the joint cumulative distribution function.  Finally the authors show some experimental results on several benchmarks and wall-clock time.

**Strengths:**

* Multi-objective optimization is an interesting topic in Bayesian optimization.
* The proposed method based on multivariate ranking seems interesting.

**Weaknesses:**

* The authors should improve writing and presentation more.  For example, figures are too small.  Also, there are some typos or grammar issues.  For example, in Theorem 2, there exist should be there exists, and in Section 3.4, Aas (2016) propose should be Aas (2016) proposes.
* Experimental results are not promising.
* According to description on experiments, the authors repeated the experiments 5 times, but variances are not reported.
* Five repetitions are not enough to validate the proposed algorithm.
* Based on the motivation, the authors argued that the use of multivariate rank and its distribution can accelerate the process of multi-objective Bayesian optimization.  However, Figure 5 does not seem to support this motivation.  The proposed methods should be faster than the other algorithms, but some results are comparable to ones of some algorithms.

**Questions:**

* In Section 2.1, I think the sentence "Often the integral is approximated by Monte Carlo ..." is not correct.  In Bayesian optimization, we often use the statistics of the posterior predictive distribution calculated directly, instead of the samples of the distribution.
* I agree that the use of ranking can be better than absolute metric values.  However, the consideration of absolute metric values is sometimes important.  What do you think of this issue?
* BOtied v1 is always worse than Botied v2 if I understand correctly.  Why do you add BOtied v1?  Is it necessary to have it?

---

### Meta-Review · Area_Chair_wGxY · 2023-12-05

**Metareview:**

This paper presents a scalable multi-objective Bayesian optimization, where the CDF indicator is proposed as a Pareto-compliant metric for evaluating the quality of approximate Pareto sets that complements the popular hypervolume indicator. All the reviewers agree that the main idea based on multivariate ranking is interesting.  However, there are a few critical concerns. Most of reviewers feel that the experimental results do not fully demonstrate what is claimed in the paper. In addition, unclarity in the writing should be improved. The authors did not address the reviewers’ concerns, without providing the rebuttal. So, all those concerns remained. We feel that the paper is not ready for being published on ICLR in its current version. Therefore, the paper is not recommended for acceptance in its current form. I hope authors found the review comments informative and can improve their paper by addressing these carefully in future submissions.

**Justification For Why Not Higher Score:**

Both writing and experiments should be much improved.

**Justification For Why Not Lower Score:**

N/A

---

### Decision · Program_Chairs · 2024-01-16

Reject